# Cannabidiol Exerts a Neuroprotective and Glia-Balancing Effect in the Subacute Phase of Stroke

**DOI:** 10.3390/ijms232112886

**Published:** 2022-10-25

**Authors:** Erika Meyer, Phillip Rieder, Davide Gobbo, Gabriella Candido, Anja Scheller, Rúbia Maria Weffort de Oliveira, Frank Kirchhoff

**Affiliations:** 1Molecular Physiology, Center for Integrative Physiology and Molecular Medicine (CIPMM), University of Saarland, D-66421 Homburg, Germany; 2Laboratory of Brain Ischemia and Neuroprotection, Department of Pharmacology and Therapeutics, State University of Maringá, Maringá 87020900, Brazil

**Keywords:** cannabidiol, stroke, microglia, astrocytes, neuroprotection, in vivo two-photon laser-scanning microscopy, Ca^2+^ signaling

## Abstract

Pharmacological agents limiting secondary tissue loss and improving functional outcomes after stroke are still limited. Cannabidiol (CBD), the major non-psychoactive component of *Cannabis sativa*, has been proposed as a neuroprotective agent against experimental cerebral ischemia. The effects of CBD mostly relate to the modulation of neuroinflammation, including glial activation. To investigate the effects of CBD on glial cells after focal ischemia in vivo, we performed time-lapse imaging of microglia and astroglial Ca^2+^ signaling in the somatosensory cortex in the subacute phase of stroke by in vivo two-photon laser-scanning microscopy using transgenic mice with microglial EGFP expression and astrocyte-specific expression of the genetically encoded Ca^2+^ sensor GCaMP3. CBD (10 mg/kg, intraperitoneally) prevented ischemia-induced neurological impairment, reducing the neurological deficit score from 2.0 ± 1.2 to 0.8 ± 0.8, and protected against neurodegeneration, as shown by the reduction (more than 70%) in Fluoro-Jade C staining (18.8 ± 7.5 to 5.3 ± 0.3). CBD reduced ischemia-induced microglial activation assessed by changes in soma area and total branch length, and exerted a balancing effect on astroglial Ca^2+^ signals. Our findings indicate that the neuroprotective effects of CBD may occur in the subacute phase of ischemia, and reinforce its strong anti-inflammatory property. Nevertheless, its mechanism of action on glial cells still requires further studies.

## 1. Introduction

Stroke is one of the most important causes of morbidity and mortality worldwide. Globally, from 1990 to 2019, the number of incident strokes and related deaths increased by 70% and 43%, respectively [1]. Stroke survivors are particularly vulnerable to secondary consequences, including sensory, motor, and cognitive impairments, as well as mood dysfunction [2]. Current pharmacological treatment of acute stroke essentially focuses on the reperfusion of salvageable, non-infarcted ischemic brain tissue, employing intravenous administration of tissue plasminogen activator (t-PA). However, only a small number of patients are suitable to receive effective thrombolytic therapy within its narrow therapeutic window (4–6 h) after stroke onset [3]. Therefore, the identification of new therapeutic strategies capable of counteracting neurodegenerative processes and functional deficits occurring after an ischemic insult is one of the major challenges in this field. Stroke triggers a robust inflammatory response, and resident microglia are the first cells sensing and responding to the damage [4]. Within minutes following injury, microglia become activated and secrete pro- and anti-inflammatory cytokines such as tumor necrosis factor α, interleukin-1β, and interleukin-1 receptor antagonist [5]. Concomitantly, astrocytes undergo significant changes in their internal Ca^2+^ concentration, which in turn mediate the reactive phenotype, namely reactive astrogliosis, one of the classic pathological hallmarks of ischemic stroke [6,7,8]. The astroglial and microglial reactive phenotypes and the subsequent inflammatory response after stroke encompass both beneficial and harmful effects, and their real contribution to the progression of stroke still remains unclear [9,10,11]. Breakdown of the blood–brain barrier takes place after an episode of cerebral ischemia, which facilitates the infiltration of peripheral leukocytes to the ischemic brain. Leukocytes further increase the inflammatory response by enhancing excitotoxicity and oxidative stress, accompanied by cell death [4]. These pathological events propagate microvascular dysfunction, edema expansion, and poor clinical outcome [12]. The complex pathophysiology of ischemic stroke may be the reason for the ineffectiveness of treatments that act only on some mechanisms of the ischemic cascade. Thus, therapies acting on multiple pathophysiological processes might offer promising results in the treatment of stroke. Cannabidiol (CBD), one of over 100 phytocannabinoids identified in *Cannabis sativa*, has received substantial research attention in the last few years due to its multimodal pharmacological profile and remarkable safety [13,14,15]. In humans, CBD is approved for treating spasticity in multiple sclerosis [16] and seizures associated with Lennox–Gastaut or Dravet syndromes in children [17]. In preclinical settings, several studies point towards neuroprotective actions for CBD in different pathology models such as Parkinson’s disease [18,19], Alzheimer’s diseases [20,21], epilepsy [22,23], multiple sclerosis [24,25,26], schizophrenia [27,28], cerebral ischemia [29,30], as well as spinal cord injury [31,32]. The neuroprotective effects of CBD in cerebral ischemic conditions have been associated with the capacity of this compound to restrain inflammatory responses mediated by inflamed microglial cells and activated astrocytes [9,33]. For example, reductions in microglia and astroglial reactivity were observed in mice treated with CBD after bilateral common carotid artery occlusion [33,34], a model of transient global cerebral ischemia. Improvements in neurological deficits and cerebral blood flow were accompanied by a reduced infarct size and microglial activation in mice treated with CBD after middle cerebral artery occlusion (MCAO) [35,36,37,38]. Only few studies examined the modulatory effects of CBD on glial cells in experimental models of ischemic stroke [9,35,38]. Although several studies have demonstrated a beneficial effect of CBD in focal cerebral ischemia, administration of CBD was performed before the MCAO [30,37,38,39]. To the best of our knowledge, our work is the first to explore the effect of CBD when administered acutely and after the onset of ischemia in adult mice. Furthermore, we investigated the effect of CBD on glial cells using an in vivo approach. For this purpose, we took advantage of transgenic mice with microglial expression of the enhanced green fluorescent protein EGFP [40] and the astrocyte-specific expression of the genetically encoded Ca^2+^ indicator GCaMP3 [41,42] to test the impact of CBD on microglial activation and astroglial Ca^2+^ activity over time, in vivo, using two-photon laser-scanning microscopy (2P-LSM).

## 2. Results

### 2.1. Ischemia-Induced Neurodegeneration and Neurological Deficit Are Reduced after Cannabidiol Treatment

To evaluate the effect of cannabidiol (CBD) on ischemia-induced neurological deficits and neurodegeneration, 12-week-old C57BL/6N wt mice were submitted to 15 min middle cerebral artery occlusion (MCAO) and treated with intraperitoneal (i.p.) injection of vehicle or CBD (10 mg/kg) at 30 min, 24 h, and 48 h after reperfusion. The effects of ischemia and CBD on neurological function were evaluated 24 h after reperfusion using the neurological deficit score (Figure 1a). To assess neurodegeneration, the number of dying neurons was quantified by Fluoro-Jade C (FJC) staining 2 days after sham or MCAO surgery in the hippocampal CA1 subfield, dentate gyrus (DG), thalamus, and striatum in both ipsi- and contralateral hemispheres (Figure 1b–e). Ischemic mice treated with vehicle presented a mean deficit score of 2 (moderate impairment), while sham-operated mice displayed normal behavior. Treating mice submitted to MCAO with CBD resulted in a significant reduction in the deficit score to about 1 (mild impairment) (Figure 1f). An increased percentage of FJC+ cells per slice was found in the brains of ischemic animals treated with vehicle compared to the brains of ischemic animals treated with CBD, reflecting the beneficial effect of CBD on MCAO-induced neurodegeneration (Figure 1g). While in the hippocampus (CA1, DG) and striatum, the numbers of degenerating neurons (FJC+ cells) showed only a beneficial trend between mice treated with vehicle and CBD (Figure 1h–j, respectively), administering CBD led to a significantly reduced number of dying neurons in the thalamus (Figure 1k).

### 2.2. Less Activated Microglia Cells in the Ischemic Brain after Cannabidiol Treatment

MCAO-induced microglial activation was assessed using the same experimental design as described above (Figure 2a) and by immunohistochemistry of Iba1 in the cortex, hippocampal CA1 subfield, and thalamus on the contra- and ipsilateral sides of the brain (Figure 2b). Microglial morphology was analyzed by semiautomatic extraction of morphological parameters using MIA software (see section “Materials and Methods” for details). In general, a strong activation of microglia was observed at the ipsilateral side of the MCAO when compared to sham surgery or to the contralateral side, and this activation was significantly reduced by the CBD treatment (Figure 2b–f). The soma area is most sensitive to the ischemic insult (Figure 2g). As expected, it appears to directly correlate with microglia hypertrophy. The total branch length as well as the number of process nodes are more variable in respect to the brain regions (Figure 2h,i).

In ischemic mice treated with CBD, the microglial soma area, total branch length, and number of nodes in CA1 from the contralateral brain side were not different from sham-operated animals (Figure 2g–i). Treatment with CBD was also able to reverse the ischemia-induced alterations of total branch length, i.e., the increase in the cortex as well as the decrease in the hippocampus and thalamus on the ipsilateral side (Figure 2h). In addition, ischemic mice treated with CBD exhibited a number of nodes in the cortex that did not differ from sham-operated mice (Figure 2i).

### 2.3. In Vivo 2P-LSM Revealed a Reduced Number of Microglia Cells with Smaller Somata in the Somatosensory Cortex of Ischemic Animals after Cannabidiol Treatment

In order to evaluate the progression of the MCAO insult and the concomitant CBD treatment on microglial activation in vivo, heterozygous mice with microglia-specific EGFP expression (TgH(CX_3_CR_1_-EGFP)) were submitted to MCAO 1 week after recovery from a cortical craniotomy and imaging window implantation. The ischemic mice were i.p. injected with vehicle or CBD at 30 min, 24 h, and 48 h after reperfusion. In vivo two-photon laser-scanning microscopy (2P-LSM) imaging was performed at different time points (Figure 3a,b), and the number of EGFP+ cells per field of view (FOV) and the microglial soma area were analyzed 2 days after reperfusion (Figure 3c–f). At baseline conditions, no differences in the number of cells and soma area could be detected between groups, indicating no microglial activation before the MCAO insult for all experimental groups. This also indicates that the implantation of the cranial window itself did not elicit an apparent microglia activation. One hour after the onset of ischemia, a larger number of microglia with increased soma area were found in the somatosensory cortex of ischemic mice treated with vehicle. Longitudinal analysis of the time-dependent morphological changes indicates that ischemia-induced microglial activation was strongly prevented by CBD treatment (Figure 3e,f). Similar outcomes were observed for cell density and soma size at 24 h after MCAO. The number of microglia cells and soma area did not differ between the sham-operated and CBD-treated ischemic mice at any recorded time point. Taken together, in vivo microglial imaging displayed a beneficial effect of CBD on microglia activation evoked by an ischemic insult.

### 2.4. Cannabidiol Balances Ischemia-Induced Alterations of Astroglial Ca^2+^ Signaling after MCAO In Vivo

Next, we investigated in vivo astroglial Ca^2+^ changes in the subacute phase of stroke in the somatosensory cortex of mice, and tested whether the treatment with CBD could modulate astroglial Ca^2+^ signaling. In order to record astroglial Ca^2+^, we took advantage of GLAST-CreERT2 knock-in mice to achieve cell-specific and time-controlled expression of the genetically encoded Ca^2+^ indicator GCaMP3 in astrocytes [41,42]. To induce reporter expression in astroglia, 7-week-old mice with a C57BL/6N background were treated with tamoxifen (100 mg/kg body weight, i.p., five times, once per day), and MCAO was performed 5 weeks later (Figure 4a,b). The mice were submitted to the same surgeries, treatment, and 2P-LSM protocol as described before (Figure 4c). The astroglial Ca^2+^ changes were recorded in the somatosensory cortex. Data processing and analysis was performed using a custom-made MATLAB-based toolbox (Figure 4d,e; see section “Materials and Methods” for details).

Within 48 h after ischemic insult, astrocytes displayed a strongly reduced area compared to baseline conditions. When mice were treated with CBD, the area of these regions of activity (ROAs) was much less affected (Figure 4f). Such a balancing effect of CBD was also observed, when the number of ROAs in the ischemic forebrain was quantified (Figure 4g). While ischemic mice displayed variable numbers of ROAs within 2 days after injury, CBD dampened this variability, and the Ca^2+^ signals were comparable to the baseline. A similar observation was made when the signal amplitude was quantified (Figure 4h), though the time course of post-injury changes were different when the number of ROAs or the respective signal amplitudes were investigated. However, the in vivo analysis of spontaneous Ca^2+^ signals revealed substantial variability, as indicated by the scatter of data points (Figure 4f–h); simultaneous treatment with CBD reduces such variations and shifts the respective physiological parameters of Ca^2+^ signals towards baseline conditions.

In summary, CBD is a potent drug that can reduce the consequences of ischemic insults evoked by MCAO and observed as motor impairment, neuronal cell death, microglia activation, and changes in astroglial Ca^2+^ signaling.

## 3. Discussion

Despite some recent advances in the treatment of ischemic stroke caused by large-vessel occlusions, such as mechanical thrombectomy, a neuroprotective pharmacological agent that reduces the consequences of stroke is still needed in the clinical treatment of patients [43]. In this study, cannabidiol (CBD (10 mg/kg, i.p. 30 min, 24, and 48 h after ischemia)) prevented neurological deficits induced by middle cerebral artery occlusion (MCAO) in mice. CBD also reduced ischemia-induced neurodegeneration and microglial activation, and influenced astroglial Ca^2+^ signaling. The results of this study agree with previous findings, showing the neuroprotective action of CBD treatment in different in vitro and in vivo models of neurodegeneration [18,21,26]. For example, CBD prevented learning deficit and cytokine expression induced by intraventricular administration of β-amyloid, an in vivo model of Alzheimer’s disease [20]. In vitro, pretreatment with CBD rescued the β-amyloid peptide-mediated long-term potentiation deficit in the CA1 area of hippocampal slices [44]. Moreover, in an animal model of multiple sclerosis, CBD treatment reduced the clinical signs and disease progression of experimental autoimmune encephalomyelitis. These effects of CBD were accompanied by decreased axonal damage, microglial activation, and T-cell proliferation [24]. Therefore, CBD is a promising candidate for the treatment of neuronal degeneration in humans. Indeed, CBD is very well tolerated, and produces adverse effects only at low incidence [45,46]. In clinical trials, the most common adverse effects reported after CBD administration included somnolence, sedation, fatigue, diarrhea, vomiting, and nausea [47,48]. It is worth mentioning that serious adverse effects following CBD treatment were observed in clinical trials with epilepsy, including severe somnolence, lethargy, increased hepatic transaminases, rash, and pneumonia. Nevertheless, these effects were related to the concomitant use of CBD with other antiepileptic drugs, including clobazam and valproate [49,50,51,52,53]. The impairment of sensorimotor and cognitive performance is a common outcome in rodents with MCAO [54,55,56]. Here, we observed neurological impairment in mice during the subacute phase after injury, i.e., 1 day after MCAO. CBD attenuated the effects of MCAO, reflected by a decrease in the neurological score, indicating an overall positive effect on functional outcomes after the injury. Consistent with our results, CBD led to an improvement in motor and neurological deficits in mice with MCAO and treated immediately before and 3 h after the occlusion. These effects of CBD were accompanied by a reduction in infarct size and an increase in cerebral blood flow [30,37,57]. In addition, CBD (5 mg/kg, i.p.) given once, 15 min after reperfusion, led to functional and sensorimotor recovery in neonatal rats submitted to MCAO [35]. Similar beneficial effects of CBD were also observed in MCAO rats treated with CBD by intracerebroventricular injection for 5 days before surgery [36]. However, mice submitted to MCAO and treated with CBD (3 mg/kg) from day 5 did not show any improvement in neurological score or motor coordination on day 14 after reperfusion. Taken together, it seems that the neuroprotective effects of CBD in cerebral ischemic insults are time-dependent, and might occur in the complex early phase of the injury. MCAO is known to cause a robust reduction in cerebral blood flow and consequent massive neuronal death. Neurodegeneration (detected by Fluoro-Jade C staining) was seen in the hippocampus, striatum, and thalamus of MCAO animals, 2 days after the insult. CBD treatment reduced the extension of neuronal loss induced by MCAO in the thalamus. Protective effects of CBD on neuronal death have been detected in the striatum and hippocampus of MCAO mice treated immediately before and 3 h after the occlusion [38], and gerbils submitted to global brain ischemia treated 5 min after reperfusion [58]. In addition, a reduction in necrotic neurons in the cortex was reported in newborn pigs submitted to hypoxic ischemic brain injury and treated with CBD (1 mg/kg, intravenous) 30 min after the insult [59]. In one study, CBD (10 mg/kg, i.p. 30 min before and 3, 24, and 48 h after ischemia) decreased neurodegeneration and normalized caspase-9 protein levels 21 days after global cerebral ischemia in mice [34]. Neuroinflammation is a critical aspect of stroke, which includes the early activation of microglia and production of cytokines and chemokines [60,61,62]. Depending on injury severity, microglia may present distinct functional and spatiotemporal profiles, which may protect or contribute to the ischemic injury evolution [63,64,65]. We observed an extensive microglial activation up to 2 days after MCAO, which extended from the ipsilateral side of the brain, including large areas in the cortex, hippocampus, striatum, and thalamus, to the contralateral hippocampus (Figure 2g–i). In line with our results, early microglial activation was also observed in regions outside of the middle cerebral artery (MCA) territory, such as the contralateral cortex and hippocampus [66]. In this sense, microglial reactivity not only indicates imminent ischemic neuronal damage, but possibly reflects subtle changes in neuronal activity outside the MCA territory. Selective neuronal loss, which refers to the death of single neurons with a preserved extracellular matrix, i.e., in the ischemic penumbra zone, is consistently associated with microglial activation in the first few days after injury [63,67,68,69]. Our results also show that microglia morphology varies across brain regions within the ipsilateral hemisphere. Microglia with robust de-ramification and amoeboid-like morphology were prominent in brain regions with intense neurodegeneration, i.e., hippocampus and thalamus, while hyper-ramified microglia with increased soma area were observed in the cortex. Our results support the idea that microglia morphology after stroke is an indicator of cerebral injury severity, involving initial increases in microglial ramification and cell body size, followed by de-ramification and ameboid-like morphology according to the progression of neurodegeneration. Cannabidiol-mediated neuroprotection after experimental cerebral ischemia has generally been related to the modulation of inflammation, including the control of microglia activation, and the toxicity exerted by these cells by producing proinflammatory mediators [34,59,70]. For example, in microglial cells challenged with lipopolysaccharide, CBD inhibited the release of proinflammatory cytokines (tumor necrosis factor-α and interleukin-1β) and also of glutamate, a non-cytokine mediator of inflammation closely related to excitotoxicity and neurodegeneration processes [71,72]. In mice submitted to 4 h of MCAO, repeated CBD treatment from day 1 after ischemia reduced the number of Iba1-positive cells expressing the pleiotropic HMGB1 protein (high-mobility group box 1) with proinflammatory function, which is released in high concentrations during the acute phase of ischemic processes [9]. A reduction in the number of Iba1-positive cells was also reported in neonatal rats submitted to a model of ischemic stroke and treated once with CBD (5 mg/kg, i.p.) 15 min after the injury [35]. Here, we also demonstrate the strong microglial activation 2 days after the onset of ischemia, and the effect of CBD thereon (Figure 2g–i). In our study, CBD treatment was able to revert ischemia-induced alterations in microglial morphological parameters in the cortex, CA1 area, and thalamus. To test the effects of CBD on microglia in vivo, we performed time-lapse imaging of microglial activity using two-photon laser-scanning microscopy (2P-LSM). Immediately after reperfusion, we observed that microglia became activated and increased their number in the ischemic penumbra. Supporting our findings, in vivo imaging of mice submitted to a cortical microhemorrhage has shown a coordinated pattern of microglia migration, where microglia within 200 µm of the injury migrated toward the lesion, leading to an increased microglia local density [73]. Moreover, in mice submitted to MCAO, it was demonstrated that many round-shaped microglia migrated to the peri-infarct area 24 h after the insult [74]. In vivo, microglia in the penumbra were found associated with blood vessels within 24 h post reperfusion. These perivascular microglia started to phagocytose endothelial cells, leading to the activation of the local endothelium, and contributing to the degradation of blood vessels, with an eventual breakdown of the blood–brain barrier [75]. Considering these findings, the inhibition of microglial activation within the first day after stroke could stabilize blood vessels in the penumbra, improving the outcomes after ischemia through an increase in blood flow. Our results show that CBD treatment decreased microglia activation after MCAO, consistent with other findings, and reinforces the strong anti-inflammatory profile of CBD in ischemic conditions. Additional to microglia, astrocytes also play an active role in the neuroinflammation process, producing complex and not yet completely understood responses to cerebral ischemia [76]. We tested whether focal ischemia would impact astroglial Ca^2+^ signaling, a characteristic form of excitability and reactivity in this cell type [77], and whether CBD treatment could modulate these signals. In our study, in vivo 2P-LSM imaging revealed that cortical astroglial Ca^2+^ signals in ischemic mice displayed a smaller area and amplitude up to 2 days after MCAO. In contrast to our findings, in vivo imaging of mice submitted to photothrombotic-induced focal cerebral ischemia (FCI) for 20 min showed an increase in frequency and amplitude of transient Ca^2+^ signals in astrocytes located in the penumbra area [78]. Moreover, permanent MCAO led to increased astroglial Ca^2+^ activity in the penumbra of aged mice, while a moderate reduction in Ca^2+^ activity at regions of interest was observed in adult mice [79]. A strong increase in intracellular Ca^2+^ in astrocytes associated with detrimental peri-infarct depolarizations was reported by Rakers and Petzold [80] in mice after permanent MCAO. It is known that astrocytes display enhanced intracellular Ca^2+^ oscillations in response to neuronal death and alarmin release, thus suggesting that the alterations in astrocytic Ca^2+^ dynamics detected in these studies may be specific to the peri-infarct zone as well as the penumbra zone. On the contrary, the somatosensory cortex (which we assessed via 2P-LSM) is not directly affected by local neuronal loss, but from the loss of ascending projections from the infarct and peri-infarct area, thus possibly explaining the reduction in cortical astroglial Ca^2+^ dynamics. If this is true, the dampening effect of CBD on astroglial Ca^2+^ alterations would therefore probably be a secondary effect mediated by CBD as a protective effect against neuronal loss. It is also important to consider methodological differences when using a permanent model of FCI or photothrombotic-induced FCI, most importantly for preventing cerebral reperfusion. To our knowledge, no study has been conducted to investigate the impact of CBD on astroglial Ca^2+^ signals after stroke. Here, the treatment with CBD balanced ischemia-induced alterations in astroglial Ca^2+^ signaling after MCAO. The effect of CBD was observed by an increase in ROA area and signal amplitude in ischemic animals in vivo (Figure 4f,h). Other studies have demonstrated an overall positive effect of CBD on astroglial activation. For example, in newborn piglets submitted to hypoxic ischemic brain injury, treatment with CBD attenuated the loss of cortical GFAP-positive cells and decreased the levels of S100B in the cerebrospinal fluid [81]. In mice submitted to global cerebral ischemia, CBD (10 mg/kg, i.p.) decreased the hippocampal reactivity of astrocytes (GFAP-positive) and total levels of GFAP 21 days after the insult [34]. In cultured astrocytes, CBD treatment decreased the ß-amyloid-induced release of proinflammatory mediators such as nitric oxide, tumor necrosis factor-α, and interleukin-1β. In the same study, CBD treatment (10 mg/kg, i.p.) for 15 days diminished the proinflammatory response and gliosis triggered by intrahippocampal injection of ß-amyloid [82]. It is important to mention, that these studies evaluated astroglial activation after brain insults at periods later than 2 days after the injury. CBD mitigates ischemia-induced neurological impairments and neurodegeneration. Importantly, we show a reduction in microglial activation in different brain regions after CBD treatment, and confirm these findings in vivo. Furthermore, we point out a possible effect of CBD on modulating astroglial Ca^2+^ signals in ischemic brains.

## 4. Conclusions

Overall, the present findings suggest that the functional and structural protective effects of cannabidiol (CBD) are closely associated with anti-inflammatory activity in the subacute phase of ischemia. Even though the mechanisms of action of CBD are not yet fully understood, our data have heuristic value to inspire further studies investigating the effect of CBD using different treatment schedules, for example, when administered for longer periods or later after the onset of ischemia. In conclusion, our data highlight the potential of CBD as a neuroprotective compound in stroke.

## 5. Materials and Methods

### 5.1. Ethics Statement

All animal procedures were carried out at the University of Saarland in strict accordance with the recommendations to European and German guidelines for the welfare of experimental animals and approved by the Saarland state’s “Landesamt für Gesundheit und Verbraucherschutz” in Saarbrücken/Germany (animal license number: 65/2013 and 36/2016).

### 5.2. Animals

Mice were housed at the animal facility of the Center for Integrative Physiology and Molecular Medicine (CIPMM) in Homburg under controlled temperature (22 ± 1 °C) and 12 h light–dark cycle, with food and autoclaved tap water ad libitum (standard autoclaved rodent diet, Ssniff Spezialdiäten, Soest, Germany). Experiments were conducted with 12- to 15-week-old male and female C57BL/6N wildtype (WT) and transgenic mice with C57BL/6N background. To image microglia in vivo, heterozygous knock-in mice (TgH(CX_3_CR_1_-EGFP)) were used (Cx3cr1^tm1Litt^, MGI: 2670351). To visualize astrocytic Ca^2+^ signals, the inducible CreERT2 DNA recombinase knock-in mouse line TgH(GLAST-CreERT2) (Slc1a3^tm1(cre/ERT2)Mgoe^, MGI:3830051) was crossed to the floxed reporter mouse line TgH(Rosa26-CAG-^fl^stop^fl^-GCaMP3-WPRE) (Gt(ROSA)26^Sortm1(CAG-GCaMP3)Dbe^, MGI:5659933). For simplification, hereafter, the mouse lines are termed CX_3_CR_1_^EGFP^ and GLAST^GCAMP3^, respectively [40,41,42].

### 5.3. Tamoxifen Induced Recombination

To induce GCaMP3 expression, 7-week-old mice were injected intraperitoneally (i.p.) with tamoxifen for 5 consecutive days (once per day, 100 mg/kg body weight) [83] (Figure 4b).

### 5.4. Cannabidiol Treatment

Cannabidiol (CBD (THC Pharma, cat. no. DWO161.207-2, Frankfurt, Germany)) was diluted with 1% Tween 80 in sterile isotonic saline (vehicle). The animals were randomly assigned to receive injections (i.p.) of CBD 10 mg/kg or vehicle 30 min, 24 h, and 48 h after surgery (Figure 1a, Figure 2a, Figure 3b and Figure 4c). The 10 mg/kg dose of CBD and the administration route were based on previous studies that reported a neuroprotective effect of CBD against cerebral ischemia in rodents [34,36]. To date, no adverse effects related to the use of this dose in mice have been reported in the literature [45]. It is important to mention that CBD is highly lipophilic and quickly crosses the blood–brain barrier, as previously demonstrated in mice and rats [84]. The number of mice used in the experimental groups are listed below (Table 1).

### 5.5. Middle Cerebral Artery Occlusion

Middle cerebral artery occlusion (MCAO) was realized in animals under inhalational anesthesia (1.5–2% isoflurane, 66% O_2_, and 33% N_2_O) and executed as described previously [85]. Briefly, the left common carotid artery (CCA) and the external carotid artery were permanently ligated with silk sutures. A silicone-coated filament (Doccol Corp, cat. No. 602156PK10Re, Sharon, MA, USA) was introduced through an arteriotomy and advanced into the right internal carotid artery until mild resistance was felt, indicating the filament reached the origin of the MCA to occlude the blood flow. After 15 min occlusion, the filament was gently withdrawn, and a suture was made around the CCA to prevent backflow through the arteriotomy [86]. After recovery from anesthesia, the mice were kept in their cages with free access to food and water and received appropriate pain treatment (analgesic and antiphlogistic agents for at least three consecutive days). In addition, the mice received 10% glucose (0.5 mL/30 g body weight, subcutaneously) as a fluid replacement after the surgeries.

### 5.6. Cortical Craniotomy

For in vivo two-photon laser-scanning microscopy (2P-LSM), a cranial window was prepared over the somatosensory cortex (3.4 mm posterior to bregma and mediolateral 1.5 mm, 3–4 mm diameter) using an engraving driller as previously described [87]. After bone removal, a coverslip was placed on the exposed brain and the edge was sealed with dental cement (RelyX^®^, 3M ESPE, Saint Paul, MN, USA). Finally, a custom-made metal holder (5 mm diameter) was placed over the coverslip and glued with dental cement onto the bone.

### 5.7. Neurological Score

Mice were evaluated for neurological deficits and motor impairment at 24 h after MCAO using the modified Bederson score system [88,89]. Mice were first held by the tail 1 m above the floor and observed for forelimb flexion. Thereafter, they were placed on plastic-coated paper and lateral pressure behind the shoulder was applied repeatedly in each direction with the tail held by hand, registering sliding behavior. Finally, mice were allowed to move around freely and observed for circling behavior. Mice were staged as listed below (Table 2; Figure 1a). Two mice presenting a score of 5 were euthanized and excluded from the experiment.

### 5.8. Immunohistochemistry

Mice were perfused (4% paraformaldehyde), and post-fixed coronal tissue sections (40 µm) at the hippocampal level (bregma −1.34 mm to −2.70 mm, Franklin and Paxinos, 1997) were collected with a Leica VT1000S vibratome (Leica Biosystems, Wetzlar, Germany) and used for immunohistochemistry of free-floating tissue sections as published previously [85]. Briefly, sections were incubated in blocking solution (0.3% Triton X-100, 5% horse serum in PBS, 1 h, room temperature (RT)) following incubation with polyclonal rabbit anti-Iba1 antibody (1:1000, overnight, 4 °C, Wako Chemicals USA, cat. no. 019-19741, Richmond, VA, USA) and incubation (2 h, RT) with anti-rabbit secondary antibody (1:1000, Alexa, Thermo Fisher Scientific, cat. no. 710369, Waltham, MA, USA) and 4′,6-diamidin-2-phenylindol (DAPI, f.c. 0.025 µg/mL, AppliChem, Darmstadt, Germany). Finally, the sections were mounted with Immu-Mount^®^TM (Thermo Fisher Scientific, Walthan, MA, USA).

### 5.9. Fluoro-Jade C Staining

To monitor MCAO-induced cell death, degenerating neurons were stained with Fluoro-Jade C (FJC) [90]. Vibratome sections, were mounted on gelatin-coated slides and dried at 50 °C for 30 min. The slides were immersed in 1% sodium hydroxide in 80% ethanol for 5 min, rinsed for 2 min in 70% EtOH, then for 2 min in distilled water, and incubated in 0.06% potassium permanganate solution for 15 min. Following a 1 min water rinse, the slides were transferred for 20 min into a 0.0001% solution of FJC (Sigma, cat. no. AG325, St. Louis, MO, USA) dissolved in 0.1% acetic acid vehicle in the dark. Thereafter, slides were washed in distilled water, air dried, and mounted with DPX (Sigma, cat. no. 44581 St. Louis, MO, USA).

### 5.10. Microscopic Analysis and Quantification on Fixed Brain Slices

Three brain sections per mouse, and at least three mice per group, were analyzed. The sections were the cortex, hippocampus, and thalamus in both brain hemispheres. Epifluorescent micrographs (FJC) were recorded using an automated slide scanner (AxioScan.Z1 equipped with the Colibri 7 LED Light Source and appropriate filters, and analyzed with ZEN1 software (Blue Edition, all Zeiss, Oberkochen, Germany). Confocal z-stacks were obtained using a laser-scanning microscope (LSM-710, Zeiss, Oberkochen, Germany) as previously described [85], processed with Fiji/ImageJ software [91], and displayed as maximum intensity projections for analyses with the semiautomatic morphological parameter extractor MIA software (provided by Prof. Bart Eggen, University of Groningen, The Netherlands).

### 5.11. Two-Photon Laser-Scanning Microscopy

Live in vivo imaging was performed using a custom-made 2P-LSM equipped with a mode-locked Ti:Sapphire laser (Vision II, Coherent, Santa Clara, CA, USA). Scanning and image acquisition were controlled by ScanImage software [92]. The setup was equipped with XY-galvanometer-based scanning mirrors (Cambridge Technology). The excitation wavelength of the laser was set at 890 nm and a 20×/1.0 water-immersion objective (W Plan-Apochromat, Carl Zeiss, Jena, Germany) was used. To minimize photodamage, the average excitation laser intensity was kept at a minimum for a sufficient signal-to-noise ratio, ranging from 30 to 50 mW depending on depth. The emitted light was detected by photomultiplier tubes (H10770PB-40, Hamamatsu Photonics, Hamamatsu, Japan). The imaging settings were selected every time equally (512 × 512 pixel, zoom 2 (256 × 256 µm^2^), frame rate 1.9 Hz).

For imaging, isoflurane anesthesia was used as described for MCAO. After sedation, the animals were fixed with a metal holder on a custom-made head restrainer under the objective. Before the imaging session was started, animals were administered 50 µL Texas Red dextran (50 mg/mL, 70 kDa; Invitrogen, lot. no. 1915861, Waltham, MA, USA) via tail vein injection for visualization of brain blood vessels. Time-lapse imaging of cortical microglia was performed by the repeated acquisition of fluorescence image stacks (15 focal planes with 2 µm axial spacing) recorded 50–70 µm below the dura mater. Subsequent image stacks were recorded every 60 s, a total of 20 stacks were acquired from a single field of view (FOV). Astroglial GCaMP3 signals were recorded on a single focal plane located in the somatosensory cortex at a depth of 50–70 µm below the dura mater. Three to four regions of interest were recorded for each mouse. Astroglial images and videos were further processed using the custom-made software. The mice were imaged before (baseline), 1 h, 24 h, and 48 h after MCAO, respectively.

### 5.12. Analyses of Ca^2+^ Imaging Data

Ca^2+^ event analysis was performed using a custom-made analysis software developed in MATLAB [93] as previously described [94]. The algorithm detects fluorescence fluctuations with respect to basal per-pixel fluorescence levels (F0), computed by fitting a polynomial curve along the temporal axis of each pixel. Event detection and analysis were based on the range projection of the normalized and detrended image stack (∆F⁄F0). Fluorescence events were detected as temporally correlated, local fluorescence peaks, and segmented into individual regions of activity (ROAs).

Before analysis, Ca^2+^ imaging data were run through a preprocessing pipeline performing de-noising using the PURE-LET algorithm [95], image registration, as well as a temporal median filter of size 3.

### 5.13. Statistical Analysis

Prism 8 software (GraphPad, San Diego, CA, USA) was used for the statistical analysis. Data were examined for assumptions of a normal distribution using the Shapiro–Wilk normality test. In case of normal distribution, data were analyzed using Student’s t-test and one- or two-way analysis of variance (ANOVA) as appropriate, followed by the Tukey’s or Sidak’s multiple comparison post hoc test. In the two-way repeated measures ANOVA, group was the between-subject factor and time (test day) was the within-subject factor. Non-parametric data were analyzed using the Kruskal–Wallis analysis of variance followed by Dunn’s multiple comparison post hoc test. The data are expressed as mean ± SD. Values of *p* ≤ 0.05 were considered statistically significant.

## Figures and Tables

**Figure 1 ijms-23-12886-f001:**
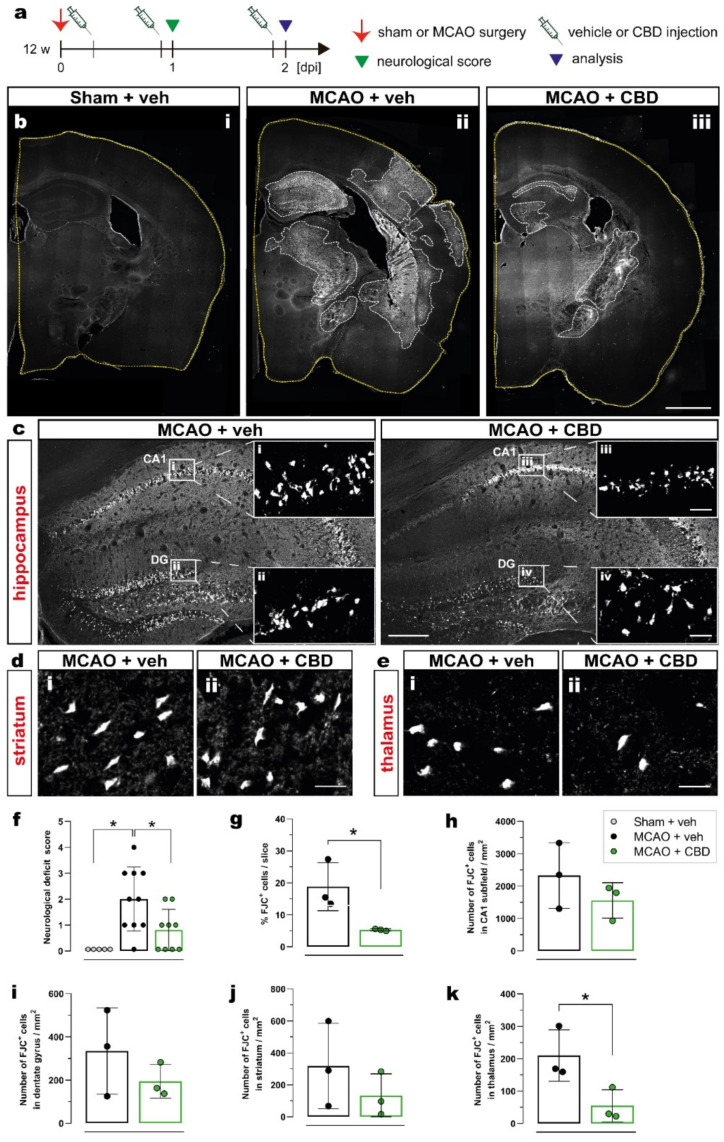
Reduction in ischemia-induced neurodegeneration and neurological deficits after CBD treatment. (**a**) Experimental design: C57BL/6N WT mice were submitted to sham or MCAO surgery and were injected with vehicle or CBD. One day after reperfusion the neurological score was determined and the brain tissue was analyzed at 2 dpi for FJC staining. (**b**) Overview of coronal brain sections among the experimental groups (i) Sham + veh, (ii) MCAO + veh, (iii) MCAO + CBD). (**c**) Overview of the hippocampus showing the FJC+ neurons; magnified view from white box in (**c**) in the CA1 area (i, iii) and DG (ii, iv). (**d**,**e**) Micrographs showing FJC+ cells in the striatum (**d**) and thalamus (**e**) of mice treated with vehicle (i) or CBD (ii). (**f**) Neurological deficit score of mice among the groups. (**g**) Quantification of ischemic area represented as percentage (%) of FJC+ cells per coronal brain slice. (**h**–**k**) Quantification of FJC+ cells represented as number of FJC+ cells in the CA1 subfield (**h**), DG (**i**), striatum (**j**), or thalamus (**k**). Data are shown as individual values (closed circles) and the means ± SD (columns and bars) of the experimental groups (*n* = 5–10/group (**f**), *n* = 3/group (**g**–**k**)). Non-parametric data were analyzed using the Kruskal–Wallis test with Dunn’s multiple comparisons test (**f**). Parametric data were analyzed using a Shapiro–Wilk normality test and were compared using Student’s unpaired t-test (**g**–**k**). * *p* ≤ 0.05. Scale bars: 1.000 µm, (**b**) overviews; 200 µm, (**c**) overviews; and 20 µm, (**d**,**e**) magnified views of (**c**). CBD, cannabidiol; DG, dentate gyrus; FJC, Fluoro-Jade C; MCAO, middle cerebral artery occlusion.

**Figure 2 ijms-23-12886-f002:**
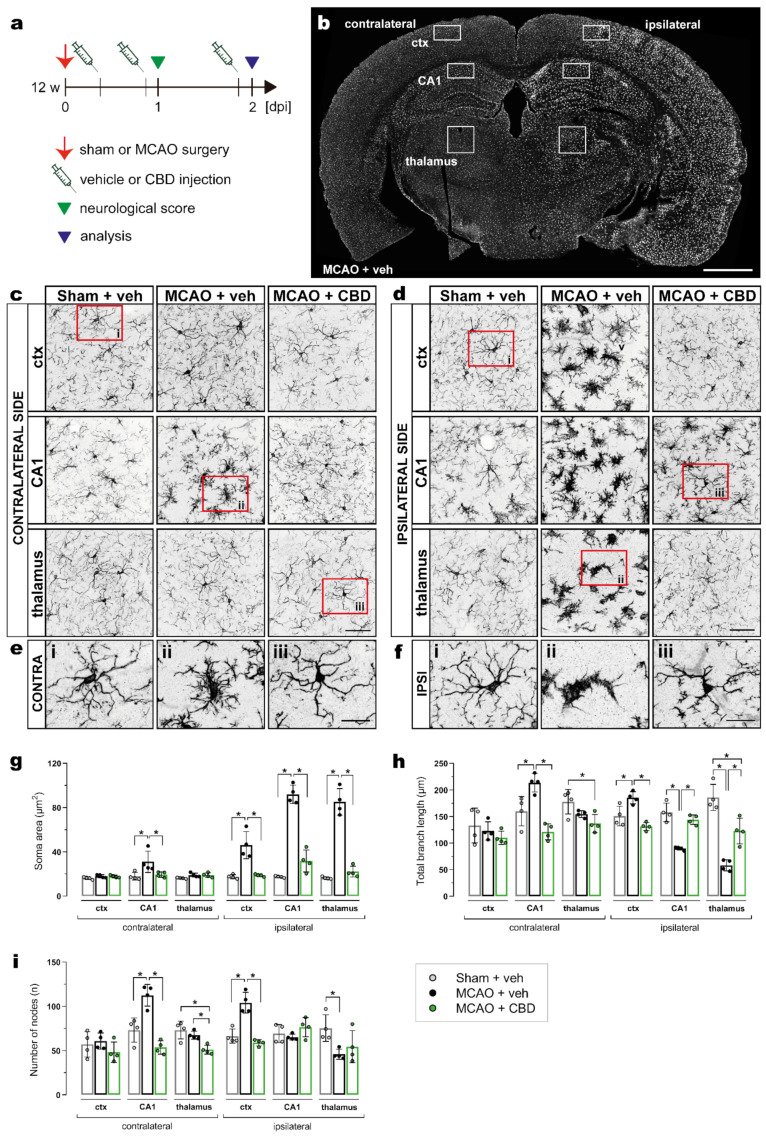
Reduction in ischemia-induced microglial activation by CBD treatment. (**a**) Experimental design: C57BL/6N WT mice were submitted to sham or MCAO surgery, and were injected with vehicle or CBD. Brain tissue was analyzed at 2 dpi for Iba-1 immunoreactivity. (**b**) Overview of coronal brain section showing the ctx, CA1 subfield, and thalamus, where the analysis for microglial morphology was performed. (**c**,**d**) Micrographs showing the heterogeneity in the morphology of Iba-1+ cells on the contralateral (**c**) and ipsilateral sides (**d**) of the brain among the groups. (**e**,**f**) Magnified views from the areas indicated by red boxes in (**c**) and (**d**) of the ctx (i), CA1 (ii), and thalamus (iii) on the contra- (**e**) and ipsilateral sides (**f**). (**g–i**) Soma area, total branch length, and number of nodes in the ctx, CA1, and thalamus on the contra- and ipsilateral sides of the brain. Data are shown as individual values (closed circles) and the means ± SD (columns and bars) of the experimental groups (*n* = 4/group). Data were analyzed using a Shapiro–Wilk normality test and were compared using an ordinary one-way ANOVA with Tukey’s multiple comparisons test. * *p* ≤ 0.05. Scale bars: 1000 µm, (**b**) overview; 50 µm, (**c**,**d**); and 20 µm, (**e**,**f**) magnified views. CBD, cannabidiol; ctx, cortex; MCAO, middle cerebral artery occlusion.

**Figure 3 ijms-23-12886-f003:**
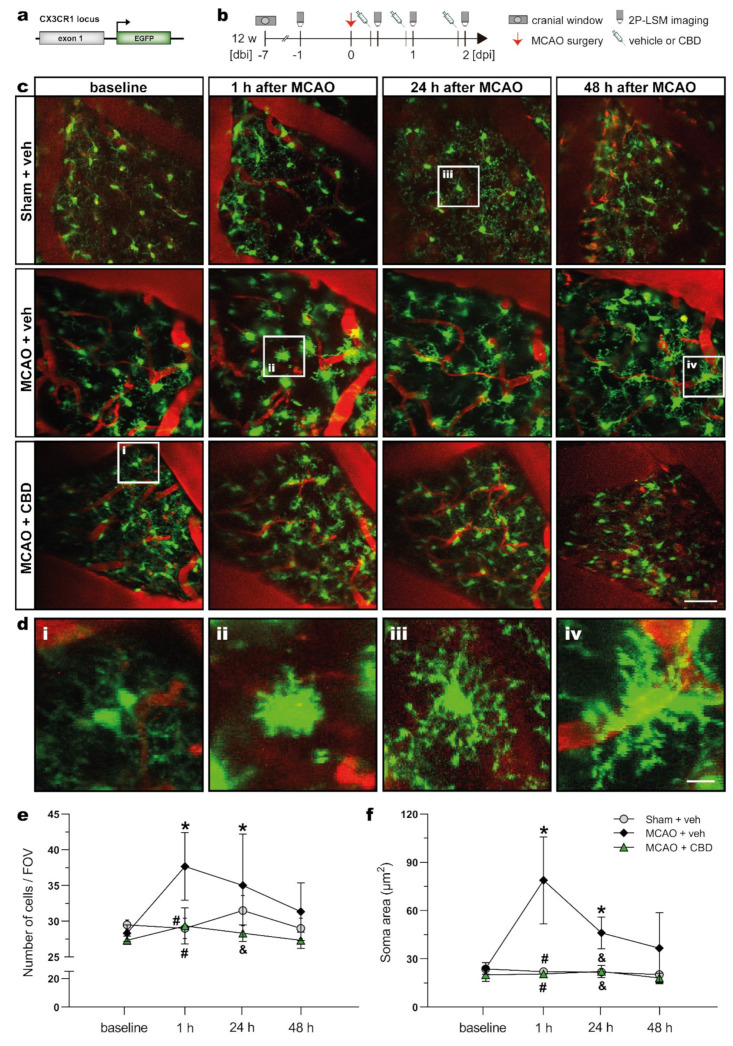
In vivo 2P-LSM demonstrated reduced microglial activation in the cortex of ischemic mice treated with CBD. (**a**) Transgene structure of mouse line used for in vivo 2P-LSM imaging. (**b**) Experimental design: CX_3_CR_1_^EGFP^ mice underwent cortical craniotomy and, after 1 week recovery, were submitted to MCAO. The ischemic mice were injected with vehicle or CBD, and 2P-LSM imaging was performed before MCAO (baseline) and 30 min after the injections. (**c**), Maximum-intensity projections of the EGFP signal for representative FOV among the experimental groups. Cerebral blood vessels were labeled with Texas Red dextran. (**d**) Magnified views indicated by white boxes in (**c**) at baseline conditions (i) and 1 h (ii), 24 h (iii), and 48 h (iv) after MCAO among the groups. (**e**) Number of EGFP+ cells found per FOV over the recorded time points. (**f**) Soma area of EGFP+ cells recorded with 2P-LSM. The bars represent the means ± SD of the experimental groups (*n* = 3/group). Data were analyzed using a Shapiro–Wilk normality test and were compared using a two-way repeated measures ANOVA with Tukey’s multiple comparisons test. Group was the between-subject factor and time (test day) was the within-subject factor. * *p* ≤ 0.05 compared to MCAO + veh group at baseline, # *p* ≤ 0.05 compared to MCAO + veh group at 1 h, and & *p* ≤ 0.05 compared to MCAO + veh group at 24 h. Scale bars: 50 µm (**c**) and 10 µm (**d**) magnified views. 2P-LSM, two-photon laser scanning microscopy; CBD, cannabidiol; EGFP, enhanced green fluorescent protein; FOV, field of view; MCAO, middle cerebral artery occlusion.

**Figure 4 ijms-23-12886-f004:**
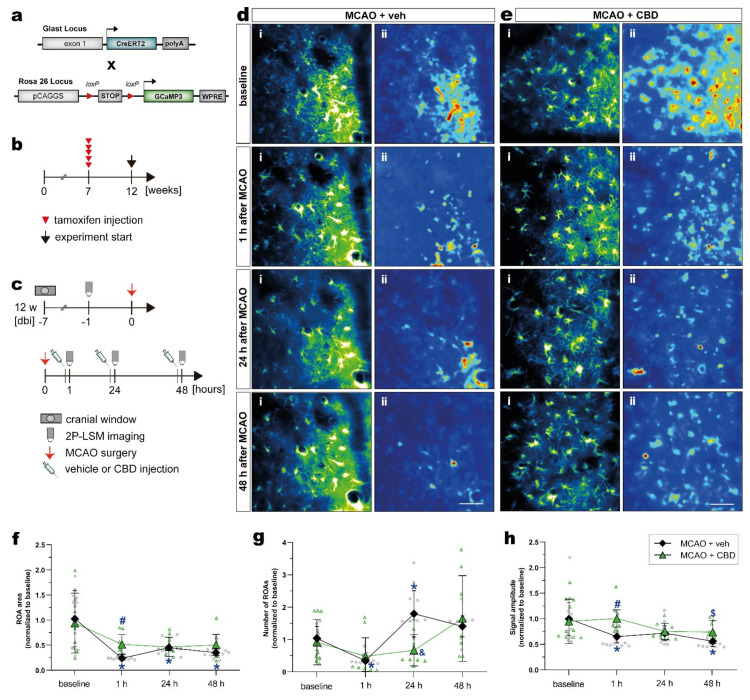
Activity-based in vivo astrocytic Ca^2+^ signals showed higher signal amplitude and ROA area in the cortex of ischemic mice treated with CBD. (**a**) Scheme of transgenic constructs carrying CreERT2 and GCaMP3. (**b**,**c**) Experimental design: GCaMP3 induction was achieved in 7-week-old mice injected with tamoxifen (**b**). GLAST^GCAMP3^ mice underwent cortical craniotomy and, after 1 week recovery, were submitted to the MCAO. The ischemic mice were injected with vehicle or CBD, and in vivo 2P-LSM imaging was performed before MCAO (baseline) and 30 min after the injections (**c**). (**d**,**e**) Automated Ca^2+^ signaling analysis for ischemic mice treated with vehicle (**d**) or CBD (**e**) using a custom-made MATLAB-based software toolbox. Maximum-intensity projections of GCaMP3 signals for representative FOV (i) over the entire recording time (up to 5 min) and relative fluorescence change projection (ii). (**f**) ROA area. (**g**) Number of ROAs. (**h**) Signal amplitude. Single datasets were analyzed using a Shapiro–Wilk normality test, and data were compared using a two-way repeated measures ANOVA with Sidak’s multiple comparisons test. Group was the between-subject factor and time (test day) was the within-subject factor. Student’s unpaired t-test was used to compare the time points between experimental groups. * *p* ≤ 0.05 compared to MCAO + veh group at baseline, # *p* ≤ 0.05 compared to MCAO + veh group at 1 h, & *p* ≤ 0.05 compared to MCAO + veh group at 24 h, and $ *p* ≤ 0.05 compared to MCAO + veh group at 48 h. Scale bars: 50 µm (**d**,**e**). 2P-LSM, two-photon laser scanning microscopy; CBD, cannabidiol; FOV, field of view; MCAO, middle cerebral artery occlusion; ROA, region of activity.

**Table 1 ijms-23-12886-t001:** Sample size according to experimental groups and analyses.

Analyses	Experimental Groups and Respective Sample Size		
	Sham + veh	MCAO + veh	MCAO + CBD
Neurological score	5	10	9
Fluoro-Jade C staining	-	3	3
Immunohistochemistry	4	4	4
Time-lapse imaging of microglia	-	3	3
Time-lapse imaging of astroglial Ca^2+^ signals	-	3	3

-: group not included in the analyses.

**Table 2 ijms-23-12886-t002:** Bederson neurological score (0–5).

Score	Bederson Neurological Score (0–5)
0	No observable deficit in motor behavior
1	Forelimb flexion
2	Forelimb flexion and decreased resistance to lateral push
3	Circling
4	Circling and spinning around the cranial–caudal axis
5	No spontaneous movement

## Data Availability

Experimental data will be provided upon reasonable request.

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
