# Peer review of "Cannabidiol Exerts a Neuroprotective and Glia-Balancing Effect in the Subacute Phase of Stroke"

_ijms, 2022, doi:10.3390/ijms232112886_

Round 1

Reviewer 1 Report

In the manuscript “Cannabidiol exerts a neuroprotective and glia-balancing effect in the subacute phase of stroke”, the authors investigated the cannabidiol effect on the cellular neurodegeneration, microglia activation, calcium microglia signaling and cognitive functions of rats in which cerebral ischemia was induced. The authors reports that astrocytes calcium signals after stroke is an unprecedent investigation in the literature.  In general, the manuscript is well-written, showing results and figures with good quality level. The introduction is able to give a good background of the research field and the importance of the investigation. My recommendation is to accept the manuscript after minor corrections, some of which are listed below.

(1) Please see the italic use in the Latin names

(2) The abstract does not contain numerical values about the results. The presence of some numbers may attract the attention of readers and highlight the significance of the results. I suggest the presence of mean ± standard deviation linked with the most expressive p values in the abstract.  

(3) use mean ± standard deviation for the description of the reduction in the deficit score in the manuscript.

(4) Figure 1 description must be improved. I recommend the description of figure 1 in sequence according to its parts. The text shows results about figures 1a, 1f, 1c-e, and others, while do not speak about Figure 1b. The same aspect may be considered in figure 2.

(5) Please see that standard error of mean (SEM) quantifies uncertainty in the estimate of the mean whereas standard deviation (SD) indicates the dispersion of the data from the mean. As readers are generally interested in knowing the variability within the sample, descriptive data should be precisely summarized with SD and not with SEM.

(6) Figure 4d and Figure 4e were not described in the text.

(7) Check the subscript use in all the text. i.e: O2 and N2O

(8) the first time in which the dose of CBD appears in the manuscript is in the discussion. I believe that this information is important in the abstract and results.

(9) The sentence “The results of this study agree with previous findings, showing the neuroprotective action of 256 CBD treatment in different in vitro and in vivo models of neurodegeneration [18, 20, 21, 257 24, 26, 55]” may be better discussed.

(10) Please add a conclusion paragraph at the discussion end and not only a sentence.

Author Response

Responses to reviewers

We acknowledge the constructive and helpful input of all reviewers. We have addressed the comments in the following point-to-point rebuttal letter and carefully revised the entire manuscript accordingly.

We believe that the current version, improved as based on the reviewers´ comments, is now suitable for publication in International Journal of Molecular Sciences.

On behalf of all co-authors and with my best regards,

Frank Kirchhoff

Reviewer #1: In the manuscript “Cannabidiol exerts a neuroprotective and glia-balancing effect in the subacute phase of stroke”, the authors investigated the cannabidiol effect on the cellular neurodegeneration, microglia activation, calcium microglia signaling and cognitive functions of rats in which cerebral ischemia was induced. The authors reports that astrocytes calcium signals after stroke is an unprecedent investigation in the literature.  In general, the manuscript is well-written, showing results and figures with good quality level. The introduction is able to give a good background of the research field and the importance of the investigation. My recommendation is to accept the manuscript after minor corrections, some of which are listed below.

  1. Please see the italic use in the Latin names.

In the revised manuscript, the Latin names are now in italics.

  1. The abstract does not contain numerical values about the results. The presence of some numbers may attract the attention of readers and highlight the significance of the results. I suggest the presence of mean ± standard deviation linked with the most expressive p values in the abstract.

Now, we included numerical values in the abstract.

  1. Use mean ± standard deviation for the description of the reduction in the deficit score in the manuscript.

Thank you for this suggestion. In the revised manuscript we presented all data with mean ± standard deviation instead mean ± SEM.

  1. Figure 1 description must be improved. I recommend the description of figure 1 in sequence according to its parts. The text shows results about figures 1a, 1f, 1c-e, and others, while do not speak about Figure 1b. The same aspect may be considered in figure 2.

In the revised manuscript, we have now described Fig. 1 and Fig. 2 in the sequence of their parts as suggested by the reviewer.

  1. Please see that standard error of mean (SEM) quantifies uncertainty in the estimate of the mean whereas standard deviation (SD) indicates the dispersion of the data from the mean. As readers are generally interested in knowing the variability within the sample, descriptive data should be precisely summarized with SD and not with SEM.

We thank the reviewer for this comment. In the revised manuscript all data are presented with mean ± standard deviation. We implemented the changes in the figure legends and description of statistics accordingly.

  1. Figure 4d and Figure 4e were not described in the text.

We thank the reviewer for this observation. We included both in the manuscript.

  1. Check the subscript use in all the text. i.e: O2 and N2O

We thank the reviewer for this correction.  

  1. The first time in which the dose of CBD appears in the manuscript is in the discussion. I believe that this information is important in the abstract and results.

We fully agree with the reviewers. This information has now been included in the abstract and results.

  1. The sentence “The results of this study agree with previous findings, showing the neuroprotective action of CBD treatment in different in vitro and in vivo models of neurodegeneration [18, 20, 21, 257 24, 26, 55]” may be better discussed.

We thank the reviewer for this critical comment. In the revised manuscript we included more information about the cited studies.

  1. Please add a conclusion paragraph at the discussion end and not only a sentence.

In the revised manuscript, we have included an extra section.

Reviewer 2 Report

The work of Meyer et al. describes the neuroprotective effects of Cannabidiol in a model of stroke. The paper is well written and the result clearly presented. Below my minor comments

ABSTRACT

- Please reduce the abstract, considering the MDPI authors' guidelines, it should be maximum 200 words.

INTRODUCTION

- Introduction must be a one big single paragraph, please unify it.

- Lines 43-44. The authors mention current pharmacological treatments but no drugs have been mentioned. Eg. antiplatelet drugs, anticoagulants and others are used in medical practice. Please add more informations about drugs.

RESULTS

- Well presented results, and excellent quality of figures. However, the figure legends are very long, please reduce them.

- Lines 154-160, these sentences should be before the related figure.

- Lines 243-248, these sentences should be before the related figure.

DISCUSSION

- Discussion must be a one big single paragraph, please unify it.

MATERIALS AND METHODS

- 4.4 The dose of Cannabidiol was based on previous studies? The route of administration?

- 4.4 The authors did a negative control with 1% Tween 80 in sterile isotonic saline (vehicle), this is very important.

- I think that the experimentals groups should be described in a table in this section, also describing the number of animals of each group.

- Line 522 "The data are expressed as mean ± SEM". I think that mean ± SD would be better.

- Please revise the abbreviations throughout the manuscript, citing the extended form the first time, and then always the abbreviation.

OVERALL

- Please add a conclusion section, indicating the translational value of this work and the future perspectives. For example, authors could test the neuroprotective activity of CBD in a model with longer timescales (es. 72h or more).

Author Response

Responses to reviewers

We acknowledge the constructive and helpful input of all reviewers. We have addressed the comments in the following point-to-point rebuttal letter and carefully revised the entire manuscript accordingly.

We believe that the current version, improved as based on the reviewers´ comments, is now suitable for publication in International Journal of Molecular Sciences.

On behalf of all co-authors and with my best regards,

Frank Kirchhoff

Reviewer #2: The work of Meyer et al. describes the neuroprotective effects of Cannabidiol in a model of stroke. The paper is well written and the result clearly presented. Below my minor comments.

  1. ABSTRACT: Please reduce the abstract, considering the MDPI authors' guidelines, it should be maximum 200 words.

We thank the reviewer for this reminder. We reduced the abstract to 200 words.

  1. INTRODUCTION: Introduction must be a one big single paragraph, please unify it.

The introduction has been changed to a single paragraph.

  1. INTRODUCTION: Lines 43-44. The authors mention current pharmacological treatments but no drugs have been mentioned. Eg. antiplatelet drugs, anticoagulants and others are used in medical practice. Please add more information about drugs.

We thank the reviewer for this comment. In the revised manuscript, we mention the current thrombolytic agent in clinical use for acute stroke therapy (i.e., tissue plasminogen activator).

  1. RESULTS: Well-presented results, and excellent quality of figures. However, the figure legends are very long, please reduce them.

We tried our best to further reduce the figure legends. We admit that we were only partially successful. The main reason is that we also try to keep figures and their legends to a certain degree understandable without cross-referencing to the main text.

  1. RESULTS: Lines 154-160, these sentences should be before the related figure.
  2. RESULTS: Lines 243-248, these sentences should be before the related figure.

We thank the reviewer for the suggestions. Now, we cited the figure after the sentences accordingly.

  1. DISCUSSION: Discussion must be a one big single paragraph, please unify it.

The discussion is now a single paragraph.

  1. MATERIALS AND METHODS: 4.4 The dose of Cannabidiol was based on previous studies? The route of administration?

The 10 mg/kg dose of CBD and administration route were based on previous studies that reported a neuroprotective effect of CBD against cerebral ischemia in rodents. We inserted this information in the manuscript including two citations.

  1. MATERIALS AND METHODS: 4.4 The authors did a negative control with 1% Tween 80 in sterile isotonic saline (vehicle), this is very important.

We thank the reviewer for this comment.

  1. MATERIALS AND METHODS: I think that the experimental groups should be described in a table in this section, also describing the number of animals of each group.

In the revised manuscript we included a table in the ‘Materials and Methods’ section describing the number of mice used in the experimental groups according to the analyses performed.

  1. MATERIALS AND METHODS: Line 522 "The data are expressed as mean ± SEM". I think that mean ± SD would be better.

In the revised manuscript, now all data are presented as mean ± standard deviation. We implemented the changes in the figure legends and statistics description accordingly.

  1. MATERIALS AND METHODS: Please revise the abbreviations throughout the manuscript, citing the extended form the first time, and then always the abbreviation.

Thank you for this comment. We have reviewed all abbreviations throughout the manuscript.

  1. OVERALL: Please add a conclusion section, indicating the translational value of this work and the future perspectives. For example, authors could test the neuroprotective activity of CBD in a model with longer timescales (es. 72h or more).

We thank the reviewer for this suggestion. We agree completely. Analyzing CBD effects over longer periods after the onset of ischemia is important and has to be the next essential to expand our findings and conclusions regarding cerebral ischemia. We are rising funds to perform such experiments in the near future. In the revised manuscript, we added a ‘Conclusion’ section as requested.

Reviewer 3 Report

I would like to thank the Authors of the present Manuscript and the Editors for the opportunity to provide commentary on this work.

To my understanding, experimentation has been performed on transgenic mice in order to evaluate the protective effects of cannabidiol (CBD), a major non-psychoactive compound of Cannabis sativa extract, on tissue loss and functional improvement in the acute phase of stroke, especially in the somatosensory cortex and in the context of microglial response. Results show that CBD can reduce microglial activation and balance calcium signals in the astroglia in the post-acute phase of stroke, thanks to its strong anti-inflammatory properties, though the specific mechanism of action involving glial cells is yet to be fully elucidated.

I want to commend the extensive experimental work that has been carried out to produce the results of this remarkable research. Although the methodology is not novel, the approach and its results are innovative and may open the doors to crucial phytopharmaceutical innovations involving anti-inflammatory CBD, the use of which is still surrounded by a cloud of judgement (and legal conundrums) in many countries around the world. Despite the complex task of describing and discussing several experimental settings, the article is well-written and easy to read. The Introduction provides sufficient background to the subject matter of stroke and CBD experimentation; the Results are well-described, as are the Materials and Methods; the Discussion section fleshes out appropriately the implications of such experimental discoveries.

I have no further suggestions to the Authors of the manuscript, as they have done a very good work that can have a relevant impact on future pharmacological and medical research.

Author Response

Responses to reviewers

We acknowledge the constructive and helpful input of all reviewers. We have addressed the comments in the following point-to-point rebuttal letter and carefully revised the entire manuscript accordingly.

We believe that the current version, improved as based on the reviewers´ comments, is now suitable for publication in International Journal of Molecular Sciences.

On behalf of all co-authors and with my best regards,

Frank Kirchhoff

Reviewer #3: I would like to thank the Authors of the present Manuscript and the Editors for the opportunity to provide commentary on this work.

To my understanding, experimentation has been performed on transgenic mice in order to evaluate the protective effects of cannabidiol (CBD), a major non-psychoactive compound of Cannabis sativa extract, on tissue loss and functional improvement in the acute phase of stroke, especially in the somatosensory cortex and in the context of microglial response. Results show that CBD can reduce microglial activation and balance calcium signals in the astroglia in the post-acute phase of stroke, thanks to its strong anti-inflammatory properties, though the specific mechanism of action involving glial cells is yet to be fully elucidated.

I want to commend the extensive experimental work that has been carried out to produce the results of this remarkable research. Although the methodology is not novel, the approach and its results are innovative and may open the doors to crucial phytopharmaceutical innovations involving anti-inflammatory CBD, the use of which is still surrounded by a cloud of judgement (and legal conundrums) in many countries around the world. Despite the complex task of describing and discussing several experimental settings, the article is well-written and easy to read. The Introduction provides sufficient background to the subject matter of stroke and CBD experimentation; the Results are well-described, as are the Materials and Methods; the Discussion section fleshes out appropriately the implications of such experimental discoveries. I have no further suggestions to the Authors of the manuscript, as they have done a very good work that can have a relevant impact on future pharmacological and medical research.

We appreciate the reviewer’s positive comments on our work. Thank you.

Reviewer 4 Report

The manuscript entitled “Cannabidiol exerts a neuroprotective and glia-balancing effect in the subacute phase of stroke” addresses the beneficial effects of cannabidiol (CBD), the major non-psychoactive component of Cannabis sativa, against experimental focal cerebral ischemia in transgenic mice and the underlying mechanisms. Interestingly, CBD mitigated ischemia-triggered neurological impairments and neuronal loss in ischemic mice. These beneficial effects were chiefly driven by ischemia-induced microglial activation alongside the balancing effect on astroglial Ca2+ signals.     

The manuscript is clearly written, and the current findings are interesting.

Comments:   

1) What is the novelty of the present work? The authors are advised to outline in the introduction section the novelty of the present work and how the study is different from previous literature that has already described the effect of cannabidiol on stroke in rodents as previously reported, for example, by Khaksar et al., 2022 (Antioxidant and anti-apoptotic effects of cannabidiol in model of ischemic stroke in rats, Brain Res Bull, 2022 Mar;180:118-130. doi: 10.1016/j.brainresbull.2022.01.001). Authors are advised to address this point and add the answers to the comment in the introduction section.

2) Did the authors detect the level of cannabidiol in the brain tissue in order to describe whether it can cross the blood-brain barrier in stroke? Does previous literature address this point previously? If yes, please add the appropriate citation in the material and methods section.  

3) How did the authors decide on the dose of cannabidiol (10 mg/kg)? How is the dose relevant to the human dose using the Human effective dose (HED) formula= animal dose x animal Km/ human Km (Nair AB, Jacob S. A simple practice guide for dose conversion between animals and humans. J Basic Clin Pharm. 2016 Mar;7(2):27-31). Authors are advised to address this point and add the answers to the comment in section 4.4.

4) The authors are advised to add the cat no. for the used chemicals and antibodies.

5) In immunohistochemistry, did the authors also perform a negative control to ensure the specific binding of antibody to the target protein? Please, add the answer to the comment in section 4.8.

6) In the statistical analysis section, given the fact that discrete variables such as neurological scores (section 4.7 and figure 1F) are non-parametric data, ANOVA analysis is not an appropriate test. The authors are advised to analyze the data using Kruskal-Wallis analysis of variance. When statistical significance is obtained, Dunn's test is applied. The authors are advised to redo the statistical analysis for the neurological score (non-parametric data) as described.

7) In the experimental design, why did not the authors incorporate an additional gp (sham + cannabidiol)? This gp may reveal any potential toxicity of the tested drug.

8) What is the LD50 for cannabidiol in mice? Is the used dose safe? Please, add the answer to the comment in section 4.4.

9) To make all figure legends stand-alone, authors are advised to add the full name of the used abbreviations at the end of each legend.

10) In the legend of figure 1, the number of animals/replicates from which data were extracted is 3-10 per group. In fact, the 3 is a low number to give reliable data for comparison. Please, specify in the figure legend which parameter was determined by 3 animals per group. Authors are advised to address this point and add the answers to the comment to all the relevant figure legends.

11) In the discussion section, authors are advised to describe the reported adverse effects of cannabidiol from the literature. 

Author Response

Responses to reviewers

We acknowledge the constructive and helpful input of all reviewers. We have addressed the comments in the following point-to-point rebuttal letter and carefully revised the entire manuscript accordingly.

We believe that the current version, improved as based on the reviewers´ comments, is now suitable for publication in International Journal of Molecular Sciences.

On behalf of all co-authors and with my best regards,

Frank Kirchhoff

Reviewer #4: The manuscript entitled “Cannabidiol exerts a neuroprotective and glia-balancing effect in the subacute phase of stroke” addresses the beneficial effects of cannabidiol (CBD), the major non-psychoactive component of Cannabis sativa, against experimental focal cerebral ischemia in transgenic mice and the underlying mechanisms. Interestingly, CBD mitigated ischemia-triggered neurological impairments and neuronal loss in ischemic mice. These beneficial effects were chiefly driven by ischemia-induced microglial activation alongside the balancing effect on astroglial Ca2+ signals. The manuscript is clearly written, and the current findings are interesting.

  1. What is the novelty of the present work? The authors are advised to outline in the introduction section the novelty of the present work and how the study is different from previous literature that has already described the effect of cannabidiol on stroke in rodents as previously reported, for example, by Khaksar et al., 2022 (Antioxidant and anti-apoptotic effects of cannabidiol in model of ischemic stroke in rats, Brain Res Bull, 2022 Mar;180:118-130. doi: 10.1016/j.brainresbull.2022.01.001). Authors are advised to address this point and add the answers to the comment in the introduction section.

Although several studies have investigated the effect of CBD in the focal cerebral ischemia model, the administration of CBD in these studies was performed before the induction of ischemia (Hayakawa et al., 2004; Mishima et al., 2005; Hayakawa et al., 2007; Hayakawa et al., 2008). For example, in the study by Khaksar and coworkers (2022), CBD was administered (i.c.v.) for five consecutive days before the MCAO. To the best of our knowledge, our work is the first to investigate the effect of CBD when administered acutely and after the onset of ischemia in adult mice. In the revised manuscript, we addressed this point in the Introduction section as requested.

  1. Did the authors detect the level of cannabidiol in the brain tissue in order to describe whether it can cross the blood-brain barrier in stroke? Does previous literature address this point previously? If yes, please add the appropriate citation in the material and methods section.

It is indeed an important comment. Yes, the pharmacokinetic profile of CBD after acute single-dose has been previously determined (intraperitoneal and oral administration in mice and rats). In the revised manuscript, we included this information and the appropriate reference on this point in the ‘material and methods’ section as requested.

  1. How did the authors decide on the dose of cannabidiol (10 mg/kg)? How is the dose relevant to the human dose using the Human effective dose (HED) formula= animal dose x animal Km/ human Km (Nair AB, Jacob S. A simple practice guide for dose conversion between animals and humans. J Basic Clin Pharm. 2016 Mar;7(2):27-31). Authors are advised to address this point and add the answers to the comment in section 4.4.

The 10 mg/kg dose of CBD and administration route were based on previous studies that reported a neuroprotective effect of CBD against cerebral ischemia in rodents. We inserted this information in the manuscript including two citations.

Concerning the question of how the dose is relevant to the human dose, the subject is very complicated. For scaling the dose by the mentioned HED empirical approach, drugs with weak hepatic metabolism, low volume of distribution, and excreted by renal route are ideal candidates. However, CBD has a high volume of distribution and is highly protein-bound (Stott et al., 2013). CBD is also strongly metabolized in the liver, with extensive involvement of the cytochrome P450 system (Zendulka et al., 2016; ÄŒerne, 2020). The main human metabolite is 7-carboxy-cannabidiol and its toxicological profile has not been investigated because experimental animals for toxicological studies (mice, rats, and dogs) do not metabolize CBD to a comparable extent as humans (Harvey et al., 1991; Ujváry & Hanuš, 2013). The primary excretion route of CBD is through faeces (84%), followed by urine (8%) (Stott et al., 2013; Gaston & Friedman, 2017). In human studies, CBD has and is being delivered in oil-based capsules or suspensions, despite its poor oral bioavailability, estimated to be 6.5% at a 3000 mg dose (Lucas et al., 2018; Lim et al., 2020). In animal studies, the most commonly route of CBD administration is intraperitoneal.

Due to all the factors mentioned above, an extrapolation of CBD dose from animals to humans is critical and could provide erroneous estimates.

Stott CG, White L, Wright S, Wilbraham D, Guy GW. A phase I study to assess the single and multiple dose pharmacokinetics of THC/CBD oromucosal spray. Eur J Clin Pharmacol 2013;69:1135–47. doi: 10.1007/s00228-012-1441-0

Zendulka O, DovrtÄ›lová G, Nosková K, et al. Cannabinoids and Cytochrome P450 Interactions. Curr Drug Metab. 2016;17(3):206-226. doi:10.2174/1389200217666151210142051

ÄŒerne K. Toxicological properties of Δ9-tetrahydrocannabinol and cannabidiol. Arh Hig Rada Toksikol. 2020;71(1):1-11. doi:10.2478/aiht-2020-71-3301

Harvey DJ, Samara E, Mechoulam R. Comparative metabolism of cannabidiol in dog, rat and man. Pharmacol Biochem Behav 1991;40:523–32. doi: 10.1016/0091-3057(91)90358-9

Ujváry I, Hanuš L. Human metabolites of cannabidiol: A review on their formation, biological activity, and relevance in therapy. Cannabis Cannabinoid Res 2016;1:90–101. doi: 10.1089/can.2015.0012

Gaston TE, Friedman D. Pharmacology of cannabinoids in the treatment of epilepsy. Epilepsy Behav. 2017;70(Pt B):313-318. doi:10.1016/j.yebeh.2016.11.016

Lucas CJ, Galettis P, Schneider J. The pharmacokinetics and the pharmacodynamics of cannabinoids. Br J Clin Pharmacol. 2018;84(11):2477-2482. doi:10.1111/bcp.13710

Lim SY, Sharan S, Woo S. Model-based analysis of cannabidiol dose-exposure relationship and bioavailability. Pharmacotherapy 2020. doi: 10.1002/phar.2377

  1. The authors are advised to add the cat no. for the used chemicals and antibodies.

The catalogue numbers have been added.

  1. In immunohistochemistry, did the authors also perform a negative control to ensure the specific binding of antibody to the target protein? Please, add the answer to the comment in section 4.8.

Immunohistochemistry for Iba-1 was also performed on brain tissue from sham-operated mice treated with vehicle (Sham+veh group). As expected, no microglial activation was observed (see Fig. 1b [i]) and therefore quantification was not performed in this group.

  1. In the statistical analysis section, given the fact that discrete variables such as neurological scores (section 4.7 and figure 1F) are non-parametric data, ANOVA analysis is not an appropriate test. The authors are advised to analyze the data using Kruskal-Wallis analysis of variance. When statistical significance is obtained, Dunn's test is applied. The authors are advised to redo the statistical analysis for the neurological score (non-parametric data) as described.

We thank the reviewer for this correction. We analyzed the data using Kruskal-Wallis test. Since statistical significance was observed, the multiple comparisons Dunn’s test was applied. In the revised manuscript, the appropriated changes were implemented in the figure legend and in the description of the ‘Statistical analysis’ section.  

  1. and 8. In the experimental design, why did not the authors incorporate an additional gp (sham + cannabidiol)? This gp may reveal any potential toxicity of the tested drug.

What is the LD50 for cannabidiol in mice? Is the used dose safe? Please, add the answer to the comment in section 4.4.

We do agree with the reviewer that an additional control group could have been included. However, given the fact that the LD50 for cannabidiol is 29.4 mg per 30 g (considering 30 g the average weight of an adult mouse) and we used 0.3 mg per 30 g of adult mouse weight )i.e. 10 mg/kg dose of CBD) which is roughly 1 % of the LD 50. Therefore, we felt confident that this dose would not generate a toxic effect in sham-operated mice. Furthermore, to our knowledge, no adverse effects related to the use of this dose in mice have been reported to date in the literature. In addition, by omitting this group we also followed the 3R of animal experimentation where we obliged to reduce experimental animal numbers as much as possible. As suggested by the reviewer, we added this information and an appropriate reference in the section 4.4 (in the revised manuscript section 5.4).

  1. To make all figure legends stand-alone, authors are advised to add the full name of the used abbreviations at the end of each legend.

We thank the reviewer for this suggestion. We included at the end of each figure legend the full name of the used abbreviations.

  1. In the legend of figure 1, the number of animals/replicates from which data were extracted is 3-10 per group. In fact, the 3 is a low number to give reliable data for comparison. Please, specify in the figure legend which parameter was determined by 3 animals per group. Authors are advised to address this point and add the answers to the comment to all the relevant figure legends.

In the revised manuscript, we specified in the figure legends which parameters were determined by 3 animals/group. Additionally, we have inserted a table in the ‘Materials and Methods’ section describing the number of mice used in the experimental groups according to the performed analyses.

  1. In the discussion section, authors are advised to describe the reported adverse effects of cannabidiol from the literature.

In the revised manuscript, we included a paragraph addressing this point in the discussion section as advised by the reviewer.

CBD is very well tolerated and, although with a low incidence, it can produce adverse effects (Huestis et al., 2019; Dos Santos et al., 2020). In clinical trial data, the most common adverse effects reported after CBD administration include somnolence, sedation, fatigue, diarrhea, vomiting and nausea (McGuire et al., 2018; Masataka et al., 2019). Serious adverse effects following CBD treatment were observed in clinical trials with epilepsy, including severe somnolence, lethargy, increased hepatic transaminases, rash and pneumonia. Importantly, these effects were related to the concomitant use of CBD with other antiepileptic drugs, including clobazam and valproate (Geffrey et al., 2015; Devinsky et al., 2017; Gaston et al., 2017; Devinsky et al., 2018; Thiele et al., 2018).

Huestis MA, Solimini R, Pichini S, Pacifici R, Carlier J, Busardò FP. Cannabidiol Adverse Effects and Toxicity. Curr Neuropharmacol. 2019;17(10):974-989. doi:10.2174/1570159X17666190603171901

Dos Santos RG, Guimarães FS, Crippa JAS, et al. Serious adverse effects of cannabidiol (CBD): a review of randomized controlled trials. Expert Opin Drug Metab Toxicol. 2020;16(6):517-526. doi:10.1080/17425255.2020.1754793

McGuire P, Robson P, Cubala WJ, et al. Cannabidiol (CBD) as an Adjunctive Therapy in Schizophrenia: A Multicenter Randomized Controlled Trial. Am J Psychiatry. 2018;175(3):225-231. doi:10.1176/appi.ajp.2017.17030325

Masataka N. Anxiolytic Effects of Repeated Cannabidiol Treatment in Teenagers With Social Anxiety Disorders. Front Psychol. 2019;10:2466. Published 2019 Nov 8. doi:10.3389/fpsyg.2019.02466

Geffrey AL, Pollack SF, Bruno PL, Thiele EA. Drug-drug interaction between clobazam and cannabidiol in children with refractory epilepsy. Epilepsia. 2015;56(8):1246-1251. doi:10.1111/epi.13060

Devinsky O, Cross JH, Wright S. Trial of Cannabidiol for Drug-Resistant Seizures in the Dravet Syndrome. N Engl J Med. 2017;377(7):699-700. doi:10.1056/NEJMc1708349

Gaston TE, Bebin EM, Cutter GR, Liu Y, Szaflarski JP; UAB CBD Program. Interactions between cannabidiol and commonly used antiepileptic drugs. Epilepsia. 2017;58(9):1586-1592. doi:10.1111/epi.13852

Devinsky O, Patel AD, Thiele EA, et al. Randomized, dose-ranging safety trial of cannabidiol in Dravet syndrome. Neurology. 2018;90(14):e1204-e1211. doi:10.1212/WNL.0000000000005254

Thiele EA, Marsh ED, French JA, et al. Cannabidiol in patients with seizures associated with Lennox-Gastaut syndrome (GWPCARE4): a randomised, double-blind, placebo-controlled phase 3 trial. Lancet. 2018;391(10125):1085-1096. doi:10.1016/S0140-6736(18)30136-3